# Stable isotope evidence for pre-colonial maize agriculture and animal management in the Bolivian Amazon

Tiago Hermenegildo[1,2] ✉, Heiko Prümers [3], Carla Jaimes Betancourt [4], Patrick Roberts [1] & Tamsin C. O'Connell [5]

Over the past decade, multidisciplinary research has seen the Amazon Basin go from a context perceived as unfavourable for food production and large-scale human societies to one of 'garden cities', domestication, and anthropogenically influenced forests and soils. Nevertheless, direct insights into human interactions with particular crops and especially animals remain scarce across this vast area. Here we present new stable carbon and nitrogen isotope data from 86 human and 68 animal remains dating between CE ~700 and 1400 from the Llanos de Mojos, Bolivia. We show evidence of human reliance on maize agriculture in the earliest phases before a reduction in the dietary importance of this crop between CE 1100 and 1400. We also provide evidence that muscovy ducks (*Cairina moschata*), the only known domesticated vertebrate in the South American lowlands, had substantial maize intake suggesting intentional feeding, or even their domestication, from as early as CE 800. Our data provide insights into human interactions with Amazonian ecosystems, including direct evidence for human management of animals in pre-colonial contexts, further enriching our understanding of human history in what was once considered a 'counterfeit paradise'.

During the past five decades, archaeological understanding of human adaptations and past subsistence in the Amazon Basin has gone through dramatic changes. Early views, drawing heavily on environmental determinism, argued that human populations could only thrive in the plentiful floodplains ('várzea') on the basis of the cultivation of manioc and fishing, while the vast interfluvial areas ('terra firme') were portrayed as having poor soils and low protein availability, limiting human societies to small and scattered groups[1–4]. Recent developments in archaeobotany have, however, presented evidence for domesticated and managed species since the early Holocene[5–8]. Domesticates such as manioc, squash, sweet potatoes and yams appear throughout the Amazon Basin between ~8000 and 5000 BCE, while maize shows a later introduction ~4500 BCE but a nearly ubiquitous presence from ~1000 BCE[9–22]. Moreover, surveys, remote sensing and Indigenous traditional knowledge have highlighted the size of pre-colonial Indigenous populations, their impact on soils (Amazonian Dark Earths) and forest species[23–25], and their organization into low-density urban societies[26–28].

In the Southwest Amazon, the Llanos de Mojos (LdM) plays a pivotal role in these discussions given the early phytolith evidence of squash and manioc dating to ~11,000 years ago[5] and the oldest known record of maize in the Amazon Basin ~4850–4500 BCE[5,29], with maize remaining particularly abundant throughout the region between CE

[1]isoTROPIC Research Group, Max Planck Institute of Geoanthropology, Jena, Germany. [2]Laboratory of Tropical Archaeology, Museum of Archaeology and Ethnology, University of São Paulo, São Paulo, Brazil. [3]German Archaeological Institute, Commission for Archaeology of Non-European Cultures, Bonn, Germany. [4]Department for the Anthropology of the Americas, University of Bonn, Bonn, Germany. [5]Department of Archaeology, University of Cambridge, Cambridge, UK. ✉e-mail: thermenegildo@gmail.com

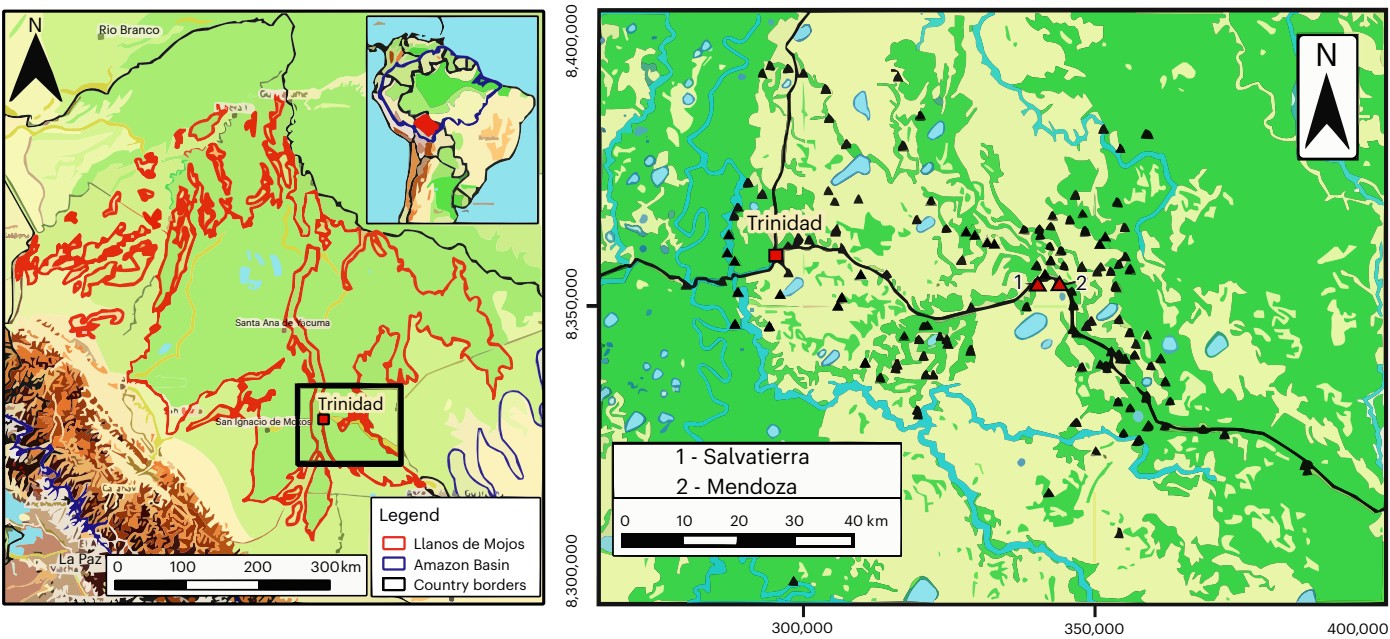

**Fig. 1 | Llanos de Mojos and the studied sites.** Left: map of Llanos de Mojos, generated using QGIS 3.34 (https://qgis.org/) with layer OpenTopoMap (https://opentopomap.org/) under CC BY-SA 3.0. Graphic by T. Hermenegildo. Right: map of the studied sites, adapted from ref. 91 Fig. 9, coordinates in UTM, zone 20S, graphic by H. Prümers.

500 and 1400 (refs. 11,13,21,22,30). Furthermore, the LdM has one of the most extensive and intricate complexes of earthworks in the South American lowlands, the Casarabe culture. Spreading for over 4,500 km² of the southeast portion of the LdM (Fig. 1), the Casarabe culture encompasses over 189 large, monumental mounds interconnected through nearly 1,000 km of canals and causeways[31]. The sheer volume of sites and their architectural layout, divided into a four-tier settlement system, ranging from large primary centres (150–300 ha) to small forest islands (~0.3 ha), indicate that the people of the Casarabe culture created a new social and public landscape through monumentality, leading to low-density urbanism[27]. The extent and complexity of the Casarabe settlement network present a unique context in the South American lowlands, even when compared to the evidence of pre-Columbian urbanism in the Upper Xingu basin[26] and Ecuador[28]. The abundance of maize remains recovered in the region[11,13,21,22,29] indicates that this crop played a key role in the emergence of the Casarabe culture and that these populations probably had well-developed maize agricultural systems[27]. Remains from a variety of other plants including manioc, sweet potatoes, squash, chili peppers, peanuts and unidentified palms have also been discovered[11,13,21,22,29,30]. Meanwhile, the discovery of muscovy duck remains in these same contexts suggests a potentially close relationship with one of the few animals to go on to be domesticated in the Neotropics[32]. Nevertheless, direct diachronic insights into human dietary reliance and interaction with different animals remain sorely lacking for this part of the Amazon[33,34].

Stable carbon ($\delta^{13}$C) and nitrogen ($\delta^{15}$N) isotope analysis of human and animal remains have a fundamental advantage over archaeobotanical and zooarchaeological approaches as they provide time-averaged insights into human dietary reliance rather than specific 'snapshots' of foods available[35]. This approach has been widely applied in archaeological studies to document dietary change across space and time, including in the context of the importance of specific crops[36–39], as well as in studies of animal domestication[36,40–42]. In tropical contexts, $\delta^{13}$C analysis has provided insights into the contributions of $C_3$ or $C_4$ biomass to consumer diets[43–46]. In the particular case of the Amazon Basin, the identification of maize consumption is made easier, since it is the only $C_4$ plant potentially eaten by humans within a biome dominated by

$C_3$ plant species with comparatively lower $\delta^{13}$C values, ranging between around −25‰ and −35‰[47]. This picture can be complicated by variable freshwater and local terrestrial Amazonian $\delta^{13}$C baselines[44,48] and the fact that capybaras have been shown to consume $C_4$ plants growing in riverine contexts[48]. The $\delta^{15}$N analysis of bone collagen can provide further insights into the amount of animal protein consumed, allowing some further discrimination of dietary patterns between direct and indirect $C_4$ consumption. Although collagen preservation has been considered unlikely in tropical contexts[49], growing application of isotope analysis and ancient DNA studies in regions such as the Amazon[50–52] is highlighting its potential to yield novel bioarchaeological data.

Despite the key role of stable isotope analysis in dietary studies, the available data for the Amazon are still limited and scattered throughout its vast territory, with only a few sites and regions investigated across an area equivalent to that of Europe[18,43,44,53–55]. Here we present stable carbon and nitrogen isotope analysis applied to human and fauna bone collagen from two sites, Salvatierra and Mendoza (Fig. 1), with monumental architecture belonging to the Casarabe culture, dated between CE ~500–1400 (see Supplement 1 for a description of sites, burials and chronologies). We use the isotopic data to examine local subsistence strategies, particularly maize's contribution to the Casarabe people's diets throughout 700 of the 900 years of mound occupation. In addition, we explore the local fauna diets, in particular muscovy ducks, and their relationship to the obtained human isotopic values. The large sample size and detailed chronology of human remains, as well as the ample archaeobotanical and zooarchaeological information available for these contexts, enable us to build new interpretations of human ecologies in this part of the Amazon Basin, including direct insights into human–animal and human–plant interactions through time.

## Results

The detailed stratigraphic and chronological contexts of sites Salvatierra[56] and Mendoza[57] have permitted the subdivision of their archaeological contexts into five occupation phases (1–5 chronologically; Supplementary Figs. 1 and 2), summarized in Harris matrices on the basis of the superposition of archaeological stratigraphy (Figs. 11–15 in ref. 55). We successfully analysed human bone remains from 86

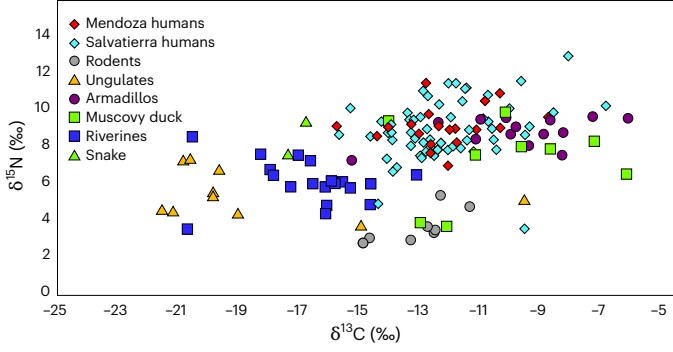

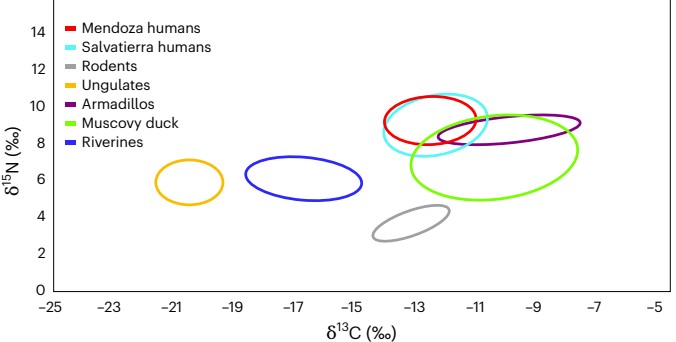

**Fig. 2 | Human and fauna stable isotope values from Salvatierra and Mendoza.** Top: δ¹³C and δ¹⁵N values from Salvatierra and Mendoza humans, and from Salvatierra fauna divided according to dietary niches. Bottom: Bayesian-inferred ellipse of the same values. Snake samples not included in Bayesian-inferred ellipse analysis as the sample size (*n* = 2) is too small to draw reliable inferred values[58]. Groups: ungulates (deer [*Mazama* sp.] and tapir [*Tapirus terrestris*]), rodents (agouti [*Dasyprocta* sp.] and capybara [*Hydrochoerus hydrochaeris*]), armadillos (*Dasypus novemcinctus* and *Euphractus sexcinctus*), muscovy duck (*Cairina moschata*), and riverines (eels [*Lepidosiren paradoxa* and *Synbranchus* spp.] and caimans [*Caiman* sp.]). The standard ellipse area of isotopic niches represents an estimated 40% of the population[58]. The standard ellipse of the ungulate group omits two outliers (results in Supplement 2) as the goal of this group is to create a reliable baseline of terrestrial C₃-consuming herbivores.

individuals from both sexes and all ages spanning phases 2–5 (Fig. 2). The faunal sample included 68 results from 11 taxa covering the earlier phases 1–3 from Salvatierra. Given their dietary niches, the fauna was divided into 5 groups: ungulates, rodents, armadillos, muscovy ducks and riverines. This grouping better represents the isotopic variability of each niche and allows for more robust statistical comparisons. All bone material showed remarkable preservation, with over 90% of sampled bone remains yielding well-preserved and uncontaminated collagen.

Faunal δ¹³C and δ¹⁵N data (detailed in Supplement 2) show a clear distinction between herbivorous C₃-consuming ungulates (mean δ¹³C −20.5 ± 1.1‰, δ¹⁵N 5.9 ± 1.1‰, *n* = 10), herbivore mixed C₃/C₄-consuming rodents (mean δ¹³C −13,1 ± 1.2‰, δ¹⁵N 3.7 ± 0.9‰, *n* = 8), mixed C₃/C₄-consuming riverine species (mean δ¹³C −16.7 ± 1.9‰, δ¹⁵N 6.1 ± 1.2‰, *n* = 19), C₄-consuming armadillos (mean δ¹³C −9.8 ± 2.3‰, δ¹⁵N 8.8 ± 0.8‰, *n* = 14) and muscovy ducks (mean δ¹³C −10.3 ± 2.6‰, δ¹⁵N 7.2 ± 2.1‰, *n* = 9). Statistical comparisons indicated significant differences among the five groups in terms of their δ¹³C values (analysis of variance (ANOVA), $F_{(4, 55)}$ = 62.81, *P* < 0.001) and δ¹⁵N values (Kruskal–Wallis $H_{(4)}$ = 37.13, *P* < 0.001), with post hoc analysis showing little relation between the groups (Supplementary Table 1). The differences in the dietary niches of faunal groups are further detailed by the Bayesian-inferred distributions (SIBER[58]), where only armadillos and muscovy ducks show any overlap of their standard ellipses (Fig. 2).

The human δ¹³C and δ¹⁵N values show similar results for both Salvatierra and Mendoza populations (respective means δ¹³C −12.3 ± 1.7‰, δ¹⁵N 9.0 ± 1.7‰, *n* = 63; and δ¹³C −12.5 ± 1.5‰, δ¹⁵N 9.3 ± 1.3‰, *n* = 23),

showing no statistical difference between both parameters (Mann–Whitney *U* tests: δ¹³C, *U* = 712.5, *Z* = −0.112, *P* = 0.91; and δ¹⁵N, *U* = 841.5, *Z* = 1.1373, *P* = 0.25). Similarly, Bayesian-inferred ellipses of the two sites show a nearly complete overlap (Fig. 2), indicating that all 86 individuals from both sites had very similar diets. This is not surprising given the proximity of the two sites (Fig. 1 and Supplementary Fig. 3) and similarities in chronology and material culture[56]. The high δ¹³C values found in all the human population indicate diets primarily based on C₄ sources throughout the 700-year occupation sequence. This is reinforced by the Bayesian inference analysis where both human ellipses show substantial overlap with the δ¹³C values of the C₄-consuming fauna (Fig. 2).

Segmenting the results from both sites on the basis of the distinct occupation phases (*n* = 73) shows the highest δ¹³C values (mean δ¹³C −10.2 ± 1.8‰, δ¹⁵N 9.7 ± 1.8‰, *n* = 10) during the early phase 2 (CE 700–800), with a gradual decrease in the subsequent phases 3 (CE 800–1100, mean δ¹³C −12.3 ± 1.3‰, δ¹⁵N 8.5 ± 1.5‰, *n* = 27), 4 (CE 1100–1350, mean δ¹³C −12.7 ± 1.6‰, δ¹⁵N 9.4 ± 1.3‰, *n* = 27) and 5 (CE 1350–1400, mean δ¹³C −12.9 ± 1.5‰, δ¹⁵N 8.3 ± 2.1‰, *n* = 9). No diachronic change is apparent in the human δ¹⁵N values (Kruskal–Wallis, $H_{(3)}$ = 6.91, *P* = 0.075). However, the δ¹³C values show a significant difference between the phases (ANOVA, $F_{(3,69)}$ = 7.40, *P* < 0.001), with further post hoc tests showing a difference between the phase 2 population and all the subsequent phases 3–5 (Supplementary Table 2). Bayesian inference reinforces the ANOVA results, showing a considerable overlap in the inferred ellipses of phases 3, 4 and 5, while phase 2 shows higher overall projected δ¹³C values and minimal overlap with other phases, particularly the later phases 4 and 5 (Fig. 3a). The exclusion of one infant individual from phase 5 (detailed in Supplement 1) with a seemingly unusual δ¹⁵N value (3.6‰) substantially alters the projected distribution for this period, reinforcing the distance between the dietary niches of phases 2 and 5 as there is no overlap between the ellipses (Fig. 3b).

## Discussion

The human stable isotope evidence from the two sites, when viewed in the light of the abundant archaeobotanical evidence of maize recovered at Salvatierra[11,13], makes it clear that maize was a vital dietary component for the Casarabe populations between CE 700 and 1400, particularly during the earliest occupation phase. The δ¹³C values of the humans from phase 2 are the highest thus far documented across the Amazon Basin[43,44,53,54], comparable to Mayan maize agriculturalist populations from the Classic and Late Classic Periods in Guatemala[59,60] as well as the earliest evidence of maize staple diets in Belize ~2000 BCE[61] (Supplementary Table 3 and Supplementary Fig. 3). The decline in δ¹³C values through time and the lack of change in δ¹⁵N values, demonstrate that from CE ~800 there was a reduction in the contribution of maize to human diets. A similar trend is also observed in Salvatierra's macrobotanical remains, where the densities of recovered maize are much higher during phases 1 and 2, double that of the subsequent phases 3 and 4/5 (ref. 11). This could either be an indication of a decrease in maize production alongside a local diversification of food production encompassing the thousands of artificial forest islands found in the region[5], or a reflection of increased trading of resources with more forested areas to the east[62–65], favoured by the extensive network of canals, causeways and rivers. Either way, future studies are needed to test these hypotheses.

The sampled muscovy duck population shows even higher overall δ¹³C values than the human population, with some individuals displaying the highest δ¹³C values (>−9‰) in this study. While capybaras have been shown to also have high δ¹³C values in the Amazon Basin due to the consumption of some wild riverine C₄ plants, this phenomenon seems to be most predominant in modern capybara, while muscovy ducks have not been documented to consume such plants in the wild[66,67]. The fauna assemblage analysed in this study shows other groups with elevated δ¹³C values (rodents, riverines and armadillos); however, their representation in the total faunal assemblage is small

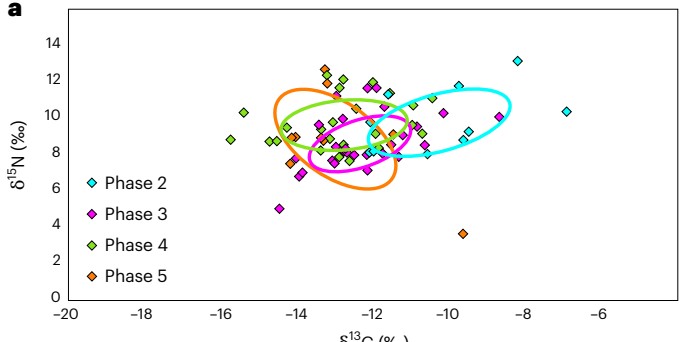

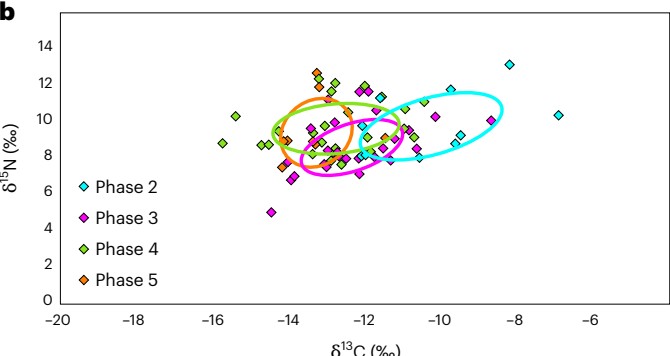

**Fig. 3 | δ¹³C and δ¹⁵N values from the Salvatierra and Mendoza populations divided according to ceramic phase. a**, Inferred ellipses for the human populations living during the different ceramic phases of Salvatierra and Mendoza. **b**, Same results excluding exceptional individual LS1218a from phase 5, showing no ellipse overlap between phases 2 and 5. The standard ellipse area of isotopic niches represents an estimated 40% of the population[58].

when compared with deer remains[68], particularly when considering their relative weights (tables 1c and 2 in ref. 68, and further discussion in Supplement 1). Furthermore, the intermediate δ¹³C values found in the riverine fauna and rodents probably reflect the local balance of $C_3$ and $C_4$ plants, while the elevated values in armadillos is a largely unknown phenomenon as other studies show variable results[48,69], including elevated δ¹³C values in pre-maize contexts[70]. This, alongside the very high δ¹³C values and the lack of change in δ¹⁵N values, suggests that muscovy ducks were consuming significant amounts of maize as local wild $C_4$ plant consumers (rodents) have significantly lower δ¹³C and δ¹⁵N values (Fig. 2 and Supplementary Table 1). Combined with later zooarchaeological evidence from Salvatierra of ducks with confinement-related pathologies[68], it is safe to suggest that muscovy ducks were actively managed at Salvatierra, intentionally fed maize since CE ~800 and kept since at least CE ~1100. Similar direct stable isotope evidence for the management of animals has not been previously reported in the Amazon Basin and indeed the entire South American lowlands. Similar isotopic evidence indicative of maize feeding practices was also reported in muscovy duck from Panama[71] (Supplementary Table 4 and Supplementary Fig. 4), suggesting that maize was a key element in the domestication of ducks throughout the American continent. Muscovy ducks are known for being the only domesticated vertebrate in all of the lowlands of South America, evident in the archaeological record and in colonial accounts of domesticated muscovy ducks in the Llanos de Mojos[72]; however, understanding of this process has remained largely unknown[32,33,66]. The data presented here provide support that humans were feeding and keeping muscovy ducks in the Bolivian Amazon from as early as CE 800 while also highlighting the role of maize in the domestication process.

The presence of maize as a dietary staple from at least CE 700 found at the Salvatierra site is an indication that maize was already well established in the region before the emergence of the Casarabe culture in CE ~500. This may support suggestions that maize agriculture had a central role in the expansion of the enormous network of settlement sites built by the Casarabe culture[27]. The evidence of maize remains dating to ~4500 BCE[5] and its later intensification into a staple crop are indications that the Llanos de Mojos region was crucial in the introduction and adaptation of maize into the Amazonian context of high heat and humidity. A recent research based on genomic, linguistic, archaeological and palaeoecological data places the southwestern Amazon as a 'secondary improvement centre'[73] (p. 1310) for partially domesticated maize since ~4500 BCE, before its expansion and divergence into other South American varieties. The authors also suggest a second major east-to-west cultural expansion of maize traditions, associated with geometric enclosures in the Upper Tapajós[74] and Upper Xingu[26] dating to CE ~800–1000. Furthermore, the predicted geographic distribution of earthworks is influenced by the sum of exchangeable base cation concentration in the surface soil[75] across the Amazon Basin, with a higher probability of earthworks in areas with higher overall soil fertility[75,76]. This wide area covers most of the southern rim of the Amazon biome, from Acre/Peru to the Xingu/Tocantins basin, hinting at a possible relationship between maize, urbanism and earthworks in the Southern Amazon.

The stable isotope data showing maize as a staple in human diets add to the abundant archaeobotanical evidence[11,13,21,22,29,30], the significance of Southwest Amazon to maize dispersion[73], and accounts of its importance to past[72,77,78] and present Indigenous peoples[79], demonstrating that maize had a far more central role in the history of Amazonian occupation than previously considered[2–4]. The data presented here show that, at least in certain contexts of the Amazon Basin, maize could have been more relevant to humans than manioc. Moreover, it may have played a critical role in the largely unknown domestication process of the muscovy duck, similar to its apparent role in the domestication and management of other animals in the Americas such as the turkey[80] and the guinea pig[81]. Future investigations in CE pre-700 contexts, including those found in the large primary centres of the settlement network such as Cotoca and Landivar, will be fundamental in understanding the history of maize in the Llanos de Mojos and the Amazon as a whole, while isotopic analyses in other areas of the Amazon Basin will probably continue to highlight the local and regional variability of human economies across this diverse region. Such a multidisciplinary approach will yield essential insights into the growing evidence for complex human, plant and now animal, interactions in the tropical rainforests of South America.

## Methods

### Sites, samples and chronologies

This research complies with the ethics guidelines for research with human remains proposed by ref. 82. The bone assemblage used in this study includes 159 samples collected at the German Archaeological Institute (Deutsches Archäologisches Institut – DAI) in Bonn, Germany, with support and permission from H. Prümers, excavation director of both sites. The excavation and export of the archaeological material was approved by the Bolivian Vice-Ministry of Culture under UNAR AUT permit number 026/02 for Mendoza and UNAR AUT permit number 019/06 for Salvatierra. After the study, all the human remains recovered were returned to Bolivia and are now part of the Kenneth Lee Museum collection in Trinidad, Beni department.

The Salvatierra and Mendoza archaeological sites are situated outside the boundaries of Indigenous Territories. The excavations were conducted on private property belonging to the Salvatierra family, with the owners' explicit consent. The local authorities of Casarabe, a relatively recent settlement dating back to the 1940s, were duly informed and expressed their support for the project. All personnel involved in the excavation were residents of Casarabe. Over the 6-year project, it became evident through discussions with the local population that the

pre-Hispanic period of the region was not regarded as part of their history, as the local population has a diverse cultural background including Mojeño, Chiquitano, Aymara and Quechua peoples. Consequently, it can be stated that no Indigenous communities participated in the fieldwork or subsequent analysis.

Human remains encompass 24 individuals from the Mendoza site covering ceramic phases 3 and 4 (CE ~800–1350), and 65 individuals from Salvatierra's phases 2 to 5 (CE ~700–1400). The age, sex and pathologies of the human skeletal remains have been described in previous studies[83,84]. The ceramic phases were determined by ref. 56 on the basis of stratigraphic evidence. Collection of human bone material focused on acquiring only the minimum quantity to produce meaningful results while also minimizing impact to the whole skeleton by selecting only non-diagnostic sections of already fragmented remains, cutting larger fragments only as a last resource. Relevant data regarding the burials are presented in Supplements 1 and 2. Faunal remains include 70 samples from 11 distinct taxa of mammals, birds, reptiles and fish; all recovered from Mound 2 (units 9 and 10) at Salvatierra. A full description of the taxa, ceramic period and stable isotope values is available in Supplement 2.

### Collagen extraction and stable isotope analysis

All of the methods described follow the standard protocols used in the Dorothy Garrod Laboratory for Isotopic Analysis, Department of Archaeology, University of Cambridge, where all sample processing took place. Bone samples, both human and faunal, were processed using a collagen extraction method adapted from refs. 85,86. Samples between 0.5 and 1.0 g were sandblasted using aluminium oxide to remove any larger contaminants and soil. Afterwards, bone material was demineralized in 8 ml of an aqueous 0.5 M hydrochloric acid (HCl) solution. Demineralization usually took 3–7 days, depending on bone density and size, with the solution changed every 48 h. Once demineralized, the material was then rinsed three times in deionized water and then gelatinized by heating the sample in a pH 3 solution in an oven at 75 °C for 48 h. The supernatant aqueous gelatinized collagen solution was removed using a 9 ml Ezee-Filter Separator from Elkay Products (60–90 μm porosity) and transferred into pre-weighed plastic tubes. Subsequently, the samples were frozen at −80 °C before being freeze-dried for 4–5 days.

Once dried, collagen test tubes were weighed once more to calculate their collagen yields, and all individual collagen samples were subsampled in triplicates of 0.7–0.9 mg when enough collagen was available and analysed by isotope ratio mass spectrometry at the Godwin Laboratory, University of Cambridge, using a Costech elemental analyser coupled to a Thermo Finnigan MAT253 mass spectrometer. The final results were calibrated using international (International Atomic Energy Agency: caffeine [$\delta^{13}C$ −27.7‰, $\delta^{15}N$ 1.0‰] and glutamic acid-USGS-40 [$\delta^{13}C$ −26.3‰, $\delta^{15}N$ −4.5‰]) and laboratory standards (nylon, alanine and bovine liver; long-term average values in Supplementary Table 5). All calibration and uncertainty in the isotopic measurements were estimated following ref. 87. Measurement precision based on check and calibration standards ($s_{srm}$) was ±0.06‰ for $\delta^{13}C$ and ±0.07‰ for $\delta^{15}N$ (d.f. = 240). Detailed values are in Supplementary Table 6. Measurement precision specific to the samples analysed in this study ($s_{rep}$) was ±0.10‰ for $\delta^{13}C$ and ±0.06‰ for $\delta^{15}N$ (d.f. = 298). Individual measurements of each triplicate analysis are displayed in Supplement 2. Measurement bias due to systematic error ($u$(bias)) was 0.15‰ for $\delta^{13}C$ and ±0.11‰ for $\delta^{15}N$. The overall measure of precision ($u(R_w)$) was calculated to be ±0.09‰ for $\delta^{13}C$ and ±0.08‰ for $\delta^{15}N$, while the standard uncertainty ($u_c$) was ±0.18‰ for $\delta^{13}C$ and ±0.14‰ for $\delta^{15}N$.

Both human and faunal bones from Mendoza and Salvatierra show remarkable collagen preservation and low contamination levels, uncharacteristic for the humid tropics. The human bone had collagen preserved in most analysed samples, with a single individual from LS not having enough for analysis (98.9% yield) and only one sample from each site showing C/N ratios outside the acceptable range[88]. In total, 86 individuals (96.6% yield) had acceptable results. The faunal samples showed equally high preservation yields, with only two samples providing insufficient collagen for the stable isotope analysis (97.1%) and two others showing signs of contamination, resulting in 66 samples with acceptable stable isotope values (94.1%).

### Statistical analyses

The choice between parametric ($t$-test, ANOVA and Tukey) and non-parametric (Mann–Whitney, Kruskal–Wallis and Dunn) analyses of datasets was determined by Shapiro–Wilk normality tests, as it is the most powerful test[89]. Results for each compared dataset are available in Supplementary Table 7. The Bayesian-based inference model SIBER (Stable Isotope Bayesian Ellipses in R) was applied to the data, as it can accurately predict the core distribution of 40% of the population with a 95% credible interval based on only 10 results[58]. Groups with $n < 10$ typically result in an underestimation of the total area of data point distribution of the population and, consequently, its niche width. All graphics and analyses were conducted in the R statistical computing programme[90] using the SIBER package[58].

### Reporting summary

Further information on research design is available in the Nature Portfolio Reporting Summary linked to this article.

## Data availability

All relevant data supporting this study are included in the article and the supporting materials.

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

## Acknowledgements

This research was funded by the Brazilian National Council for Scientific and Technological Development (CNPq), grant number 200179/2009-8, to T.H. We thank C. Kneale (McDonald Institute for Archaeological Research, University of Cambridge) for help in sample preparation and analysis; M. Hall and J. Rolfe (Godwin Lab, Department of Earth Sciences, University of Cambridge) for help with isotopic analyses; I. Trautmann, M. Trautmann (A und O - Anthropologie und Osteoarchäologie) and J. Wahl (University of Tuebingen) for the osteological analysis of the human remains; M. Kuijpers, C. Popa and J. Watling for feedback on earlier versions of the paper.

## Author contributions

T.H. designed the research, prepared the samples and analysed the data. H.P. and C.J.B. excavated and interpreted the chronology of the sites. T.C.O. funded the stable isotope analysis. T.H., H.P., C.J.P., P.R. and T.C.O. wrote the paper.

## Funding

## Competing interests

The authors declare no competing interests.

## Additional information

**Correspondence and requests for materials** should be addressed to Tiago Hermenegildo.

# Reporting Summary

## Statistics

For all statistical analyses, confirm that the following items are present in the figure legend, table legend, main text, or Methods section.

| n/a | Confirmed | |
|---|---|---|
| ☐ | ☒ | The exact sample size ($n$) for each experimental group/condition, given as a discrete number and unit of measurement |
| ☐ | ☒ | A statement on whether measurements were taken from distinct samples or whether the same sample was measured repeatedly |
| ☐ | ☒ | The statistical test(s) used AND whether they are one- or two-sided<br>*Only common tests should be described solely by name; describe more complex techniques in the Methods section.* |
| ☐ | ☒ | A description of all covariates tested |
| ☐ | ☒ | A description of any assumptions or corrections, such as tests of normality and adjustment for multiple comparisons |
| ☐ | ☒ | A full description of the statistical parameters including central tendency (e.g. means) or other basic estimates (e.g. regression coefficient) AND variation (e.g. standard deviation) or associated estimates of uncertainty (e.g. confidence intervals) |
| ☐ | ☒ | For null hypothesis testing, the test statistic (e.g. $F$, $t$, $r$) with confidence intervals, effect sizes, degrees of freedom and $P$ value noted<br>*Give P values as exact values whenever suitable.* |
| ☒ | ☐ | For Bayesian analysis, information on the choice of priors and Markov chain Monte Carlo settings |
| ☒ | ☐ | For hierarchical and complex designs, identification of the appropriate level for tests and full reporting of outcomes |
| ☒ | ☐ | Estimates of effect sizes (e.g. Cohen's $d$, Pearson's $r$), indicating how they were calculated |

*Our web collection on statistics for biologists contains articles on many of the points above.*

## Software and code

Policy information about availability of computer code

| Data collection | Thermo Scientific Mass Spectrometry Software |
|---|---|
| Data analysis | R Core Team (2021); package SIBER (Jackson et al. 2011) |

For manuscripts utilizing custom algorithms or software that are central to the research but not yet described in published literature, software must be made available to editors and reviewers. We strongly encourage code deposition in a community repository (e.g. GitHub). See the Nature Portfolio guidelines for submitting code & software for further information.

## Data

Policy information about availability of data

All manuscripts must include a data availability statement. This statement should provide the following information, where applicable:

- Accession codes, unique identifiers, or web links for publicly available datasets
- A description of any restrictions on data availability
- For clinical datasets or third party data, please ensure that the statement adheres to our policy

All relevant data supporting this study are included in the article and the supporting materials

# Research involving human participants, their data, or biological material

Policy information about studies with [human participants or human data](). See also policy information about [sex, gender (identity/presentation), and sexual orientation]() and [race, ethnicity and racism]().

| | |
|---|---|
| Reporting on sex and gender | N/A |
| Reporting on race, ethnicity, or other socially relevant groupings | N/A |
| Population characteristics | N/A |
| Recruitment | N/A |
| Ethics oversight | N/A |

Note that full information on the approval of the study protocol must also be provided in the manuscript.

# Field-specific reporting

Please select the one below that is the best fit for your research. If you are not sure, read the appropriate sections before making your selection.

☐ Life sciences   ☒ Behavioural & social sciences   ☐ Ecological, evolutionary & environmental sciences

For a reference copy of the document with all sections, see [nature.com/documents/nr-reporting-summary-flat.pdf]()

# Behavioural & social sciences study design

All studies must disclose on these points even when the disclosure is negative.

| | |
|---|---|
| Study description | We studied the dietary composition of past human populations from the Amazon basin (Llanos de Mojos, Bolivia) based on quantitative stable isotope data ($\delta 13C$ and $\delta 15N$) obtained from bone collagen. |
| Research sample | Samples include human and faunal bone remains recovered from two monumental mound archaeological sites, Salvatierra and Mendoza, dated to around 700-1400 CE. These sites were chosen as the have most detailed chronology and the largest representation of archaeological human remains recovered in the Llanos de Mojos region to date.<br>We analysed 159 bone samples of around 1g each. Human remains include individuals of all ages (0 - 65+) consiting of 24 individuals from Mendoza and 65 from Salvatierra.<br>Fauna had 70 samples from Salvatirerra collected for analysis, encompassing eleven distinct taxa of mammals, birds, reptiles and fish. |
| Sampling strategy | Sampling strategy was based on convenience, relying on archaeological bone material from the collection of the Commission for Archaeology of Non-European Cultures of the German Archaeologial Institute (KAAK-DAI) in Bonn, Germany. Currently the collection has been returned to Bolivia (Museo Kenneth Lee, Trinidad, Beni)<br>We collected human remains from all individuals available at the time of collection.<br>Fauna samples focused on covering the most representative taxa recovered at Salvatierra (around n=10 for each) in order to provide a baseline to which the human data can be interpreted. Fauna from Mendoza was not available for analysis. |
| Data collection | Archaeological bone material was excavated by Heiko Prumers (PI), Carla Jaimes Betancourt and team.<br>Bone collagen extraction and sample preparation was conducted by Tiago Hermenegildo at the Dorothy Garrod Laboratory for Isotopic Analysis, Department of Archaeology, University of Cambridge<br>The stable isotope analysis was carried out by Catherine Kneale, Mike Hall and James Rolfe at the Godwin Laboratory, Department of Earth Sciences, University of Cambridge. |
| Timing | Stable isotope analysis took two years. The analysis or experiments are not time dependent. |
| Data exclusions | Two deer samples (SAF50 and SAF56 in S2) were not included in the Baysian inference in Figure 2 and other statistical comparisons. These samples showed unusual $\delta 13 C$ values indicating they were not C3 consumers . Further detail in S1 |
| Non-participation | No participants were involved in this study |
| Randomization | Fauna groups were defined based on taxa (sometimes only to the level of genus since remains are often fragmented)<br>Human groups were divided according to occupation phases defined by the material culture (ceramic remains) |

# Reporting for specific materials, systems and methods

We require information from authors about some types of materials, experimental systems and methods used in many studies. Here, indicate whether each material, system or method listed is relevant to your study. If you are not sure if a list item applies to your research, read the appropriate section before selecting a response.

## Materials & experimental systems

| n/a | Involved in the study |
|-----|-----------------------|
| ☒ | ☐ Antibodies |
| ☒ | ☐ Eukaryotic cell lines |
| ☐ | ☒ Palaeontology and archaeology |
| ☒ | ☐ Animals and other organisms |
| ☒ | ☐ Clinical data |
| ☒ | ☐ Dual use research of concern |
| ☒ | ☐ Plants |

## Methods

| n/a | Involved in the study |
|-----|-----------------------|
| ☒ | ☐ ChIP-seq |
| ☒ | ☐ Flow cytometry |
| ☒ | ☐ MRI-based neuroimaging |

## Palaeontology and Archaeology

**Specimen provenance**
Salvatierra and Mendoza sites, Casarabe, Beni  Department, Bolivia

**Specimen deposition**
Museo Kenneth Lee (Archaeological Museum), Trinidad, Beni, Bolivia

**Dating methods**
No new dates are provided

☒ Tick this box to confirm that the raw and calibrated dates are available in the paper or in Supplementary Information.

**Ethics oversight**
The excavation and export of the archaeological material were approved by the Bolivian Vice-Ministry of Culture under UNAR AUT permit N° 026/02 for Mendoza and UNAR AUT permit N° 019/06 for Salvatierra.

Note that full information on the approval of the study protocol must also be provided in the manuscript.

## Plants

**Seed stocks**
*Report on the source of all seed stocks or other plant material used. If applicable, state the seed stock centre and catalogue number. If plant specimens were collected from the field, describe the collection location, date and sampling procedures.*

**Novel plant genotypes**
*Describe the methods by which all novel plant genotypes were produced. This includes those generated by transgenic approaches, gene editing, chemical/radiation-based mutagenesis and hybridization. For transgenic lines, describe the transformation method, the number of independent lines analyzed and the generation upon which experiments were performed. For gene-edited lines, describe the editor used, the endogenous sequence targeted for editing, the targeting guide RNA sequence (if applicable) and how the editor was applied.*

**Authentication**
*Describe any authentication procedures for each seed stock used or novel genotype generated. Describe any experiments used to assess the effect of a mutation and, where applicable, how potential secondary effects (e.g. second site T-DNA insertions, mosiacism, off-target gene editing) were examined.*

