## [Peer review File · Nature Human Behaviour]

Stable isotope evidence for pre-colonial maize agriculture and animal management in the Bolivian Amazon

Corresponding Author: Dr Tiago Hermenegildo

This manuscript has been previously reviewed at another journal. This document only contains reviewer comments, rebuttal and decision letters for versions considered at Nature Human Behaviour.

Version 0:

Decision Letter:

23rd April 2024

Dear Dr Hermenegildo,

Thank you once again for your manuscript, entitled "Direct evidence of pre-colonial maize agriculture and animal management in the Bolivian Amazon," and for your patience during the peer review process.

Your manuscript has now been evaluated by 3 reviewers, whose comments are included at the end of this letter. Although the reviewers find your work to be of interest, they also raise some important concerns. We are interested in the possibility of publishing your study in Nature Human Behaviour, but would like to consider your response to these concerns in the form of a revised manuscript before we make a decision on publication.

In sum, we invite you to revise your manuscript taking into account all reviewer and editor comments. We are committed to providing a fair and constructive peer-review process. Do not hesitate to contact us if there are specific requests from the reviewers that you believe are technically impossible or unlikely to yield a meaningful outcome.

We hope to receive your revised manuscript within two months. I would be grateful if you could contact us as soon as possible if you foresee difficulties with meeting this target resubmission date.

- Include a "Response to the editors and reviewers" document detailing, point-by-point, how you addressed each editor and referee comment. If no action was taken to address a point, you must provide a compelling argument. When formatting this document, please respond to each reviewer comment individually, including the full text of the reviewer comment verbatim followed by your response to the individual point. This response will be used by the editors to evaluate your revision and sent back to the reviewers along with the revised manuscript.
- Highlight all changes made to your manuscript or provide us with a version that tracks changes.

Link Redacted

We look forward to seeing the revised manuscript and thank you for the opportunity to review your work. Please do not hesitate to contact me if you have any questions or would like to discuss these revisions further.

Sincerely,

[REDACTED]

Reviewer expertise:

Reviewer #1: Amazonian archaeology

Reviewer #2: Phytoliths, stable isotope analysis, archaeobotany, agriculture, Amazonian archaeology

Reviewer #3: Amazonian archaeology, land use, phytoliths

REVIEWER COMMENTS:

Reviewer #1:

Remarks to the Author:

This is an important paper on agricultural development and animal management in a major region of the Amazon Basin. The maize data are interesting and the muscovy duck data and interpretations are novel and important. I'm not an isotope expert and leave it to those who are to provide an assessment of the isotope data and their interpretations by the authors.

Reviewer #2:

Remarks to the Author:

This excellent manuscript represents a major and long awaited contribution to Amazonian archaeology and beyond that surely deserves to be published in Nature Human Behaviour.

Please find below a few minor suggestions for the authors:

I. 21: I would include the number of human skeletal remains analysed.

I. 39-40: The sentence confuses the spatial extent of crops such as manioc, squash, etc., with the timing of maize. Please clarify the timing of the appearance of these other crops. Iriarte et al. 2021 summarises them.

I. 43: 'Garden cities' is acceptable for the abstract, but in the text, I would define these low-density urban societies in more specific terms and avoid vague terms such as 'Garden cities'.

I. 48: The Casarabe culture's architectural elements should be defined more detailed beyond 'extensive and intricate complexes'. A four-tiered settlement system should be briefly mentioned.

I. 58: I would include the full list all the plants, including yams, chilli peppers, and palms.

I. 162: Why not compare with some of the most known case studies of the adoption of maize, like the Mississippian ~1000AD, the Belizian Maya (Kenneth et al. 2020), and the other sites in the Amazon? It would be helpful to include a graph displaying all the Amazonian human skeletons' carbon and nitrogen isotope analysis for reference and clear comparisons.

I. 176: The reader would like to know if you have capybara in the faunal record and, if so, what the implications are.

I. 209: Include a citation after previously considered. The role of maize in Amazonian archaeology has a long history of debate, and this paper should summarise at least some of its aspects.

Reviewer #3:

Remarks to the Author:

In "Direct evidence of pre-colonial maize agriculture and animal management in the Bolivian Amazon" the authors report the results of carbon and nitrogen stable isotopes analysis performed on collagen of human and faunal bone remains from two late Holocene archaeological sites located in the Bolivian Amazon. The study is supported by a remarkable number of samples (70-86 human samples and 60-68 faunal samples) and correct use of established laboratory and statistical techniques. Overall, the results demonstrate that the human diet in these sites heavily relied on C4 plants (likely maize) and tentatively show the potential role that maize played in the diet of Muscovy ducks, the only known domesticated vertebrate in the lowlands of South America. The aim of the study is of high relevance for the interdisciplinary community of scientists studying the intricate human-environment relationships that took place in the Amazon basin during pre-colonial times. However, while the study sheds light on the importance of maize in the diets of pre-colonial people in southwestern Amazonia, the conclusions reached by the authors regarding animal management practices and particularly the feeding of Muscovy ducks seem inadequately supported by the data provided in the article.

Please, find below major and minor issues that I encountered while reviewing your manuscript.

Major issues

1) The title of the manuscript indicates that it has “direct evidence of pre-colonial maize agriculture” and I think that it is misleading. The results of the study either provide indirect evidence of pre-colonial maize agriculture or direct evidence of pre-colonial maize consumption, given that the stable isotopes analysis was performed on bone collagen and not on agricultural soil samples (see, for example, <https://doi.org/10.1016/j.jas.2010.06.016>). However, it is true that throughout the text the results are interpreted and discussed in terms of dietary patterns and not agricultural practices. For this reason, I would suggest modifying the title to better represent the scope and content of the manuscript.

2) The study discusses potential animal management in the studied sites (see, for example, title and lines 26-28) on the basis of a) the presence of Muscovy duck remains, one of the few animals domesticated in pre-colonial South America; b) the identification of individuals with confinement-related pathologies (lines 178-179); and c) the high values of $\delta^{13}C$ detected in the bone collagen of Muscovy ducks, which could suggest that they were fed with maize (lines 177-178). However, there are some critical aspects that require further discussion and clarification. First, as shown in Figure 2, rodents, Muscovy ducks and armadillos have a similar $\delta^{13}C$ range of values when compared to the values in humans, but only the results obtained from capybaras are discussed (lines 173-177) while the ones from armadillos and other rodents are ignored. This is key if the results obtained from the ducks are going to be interpreted as the effect of maize consumption. Second, the confinement-related pathologies are only reported in bones from phase 4 in Salvatierra (see pages 361-362 in von den Driesch & Hutterer 2011, which you cite), while the Muscovy duck bones analyzed in this study come from phases 1, 2, and mainly 3. Third, file S2 contains the results of 11 Muscovy duck samples, but in the study only the results of 9 samples are included. Interestingly, only the coracoid samples which have lower $\delta^{13}C$ values were excluded but I could not find the reason neither in the main text nor in S1 and S2. Fourth, the sample size of Muscovy duck is rather low (9-11 bones) compared to the total population of bones recovered in Salvatierra (according to von den Driesch & Hutterer 2011, 89 fragments were recovered in phase 3, 283 in phase 4, and 47 in phase 5) and mostly focused on phase 3, so I wonder if the sampling strategy could have had any effect in the results obtained.

Minor issues

1) The number of samples does not seem to be consistent throughout the text, something that should be justified. In lines 100-101, 86 human bone remains and 68 faunal remains are reported. In lines 107-112, the total number of faunal remains is 60, while according to file S2 it should sum 68 (or 66, considering that 2 Muscovy duck samples were excluded too). In lines 138-142, the total number of human remains is 70, but why is not 86? Also, in related Table S2, the number of human samples sum 73 instead of 70 or 86.

2) Sugiyama et al. (2020, <https://doi.org/10.1016/j.jaa.2020.101195>) reported similar results and interpretations to yours but using samples from pre-colonial Panamanian sites. I think that the study should be cited and discussed in your manuscript.

3) In Figure 2, snakes are excluded in the bottom plot, please explain the reason in the caption and/or somewhere else.

4) Authors should review the manuscript paying careful attention to formatting details. Please, find below some issues that should be solved:

- Line 109. Delete repeated percentage symbols.
- Lines 113-114. To standardize, add comma after the name of the test (Kruskal-Wallis) and a space between the H value and the p value ($H(4)=37.13, p<0.001$).
- Lines 120-127. Italicize scientific names in the figure caption.
- Line 143. Replace “:” with “,” after Kruskal-Wallis.
- Line 144. Add name of the test to the parenthesis (ANOVA).
- Line 233. Replace “Stable” with “stable”.
- Line 461. You might remove “The authors declare no conflicts of interest” from the acknowledgements section, considering that a similar statement is provided in the competing interests section.
- Sometimes you refer to supplementary files as, for example, S1 and sometimes as Supplement 1. Please standardize.
- Lines 182-185 and 203-204 in S1. Adjust alignment.
- Line 183 in S1. Remove the bold font in $\alpha=0.05$.
- Line 193 in S1. I recommend describing the acronym SRMs for inexperienced readers (Standard Reference Materials).
- Line 204 in S1. Please standardize the use of spaces in ME=Mendoza site and SAL=Salvatierra site.

Version 1:

Decision Letter:

Our ref: NATHUMBEHAV-24030999A

5th August 2024

Dear Dr. Hermenegildo,

Thank you for submitting your revised manuscript “Stable isotope evidence for pre-colonial maize agriculture and animal management in the Bolivian Amazon” (NATHUMBEHAV-24030999A). It has now been seen by the original referees and their comments are below. As you can see, the reviewers find that the paper has improved in revision. We will therefore be happy in principle to publish it in *Nature Human Behaviour*, pending minor revisions to satisfy the referees’ final requests and to comply with our editorial and formatting guidelines, including our requirement that the excavations, export and analyses of archaeological materials all have appropriate permissions in place.

We are now performing detailed checks on your paper and will send you a checklist detailing our editorial and formatting requirements within two weeks. Please do not upload the final materials and make any revisions until you receive this additional information from us.

Sincerely,

[REDACTED]

Reviewer #3 (Remarks to the Author):

Thank you for having addressed my previous concerns and doubts in detail. I would recommend the manuscript for publication, provided that the following few minor issues are addressed:

1. Von den Driesch & Hutter (2011) analyzed samples from phases 3-5 while you analyzed samples from phases 1-3, however Driesch & Hutter (2011) detected confinement-related pathologies only in phase 4 but not in phase 3 from which most of your Muscovy duck samples come from. This may imply, regardless of the high d13C values in all Muscovy duck samples, that ducks were managed, if so, in different ways (and not that "muscovy ducks were kept and intentionally fed maize at Salvatierra from around 800 CE", lines 189-190). To reinforce the maize consumption argument, I think that it might be good to clarify this in the text (maybe in lines 187-190?) and I would also add, if available, what are the expected d13C values of (Muscovy) ducks if grown in the wild (which should have much lower d13C than the ones that you obtained).
2. In line 25, I would replace the SA Lowlands acronym for South American Lowlands (or lowlands).
3. Line 41. Remove space between BCE and 9-22.
4. Sentence in lines 183-186 is unclear to me because the low representation of groups with elevated d13C values in the total assemblage of faunal remains seems irrelevant with respect to the interpretation of such values in groups other than the Muscovy ducks. Here I would instead add your explanation to the values obtained from armadillos (that "their stable isotope values vary considerably between a variety of studies" and that "reasons are unknown [...]") and also give more context to the values obtained in rodents.
5. Line 220. Replace SW with Southwest?
6. Line 244 in supplementary file 1 seems incomplete.

Version 2:

Decision Letter:

Dear Dr Hermenegildo,

We are pleased to inform you that your Article "Stable isotope evidence for pre-colonial maize agriculture and animal management in the Bolivian Amazon", has now been accepted for publication in *Nature Human Behaviour*.

Please note that *Nature Human Behaviour* is a Transformative Journal (TJ). Authors may publish their research with us through the traditional subscription access route or make their paper immediately open access through payment of an article-processing charge (APC). Authors will not be required to make a final decision about access to their article until it has been accepted. [Find out more about Transformative Journals](https://www.springernature.com/gp/open-research/transformative-journals)

Authors may need to take specific actions to achieve [compliance with funder and institutional open access mandates](https://www.springernature.com/gp/open-research/funding/policy-compliance-faqs). If your research is supported by a funder that requires immediate open access (e.g. according to [Plan S principles](https://www.springernature.com/gp/open-research/plan-s-compliance)) then you should select the gold OA route, and we will direct you to the compliant route where possible. For authors selecting the subscription publication route, the journal's standard licensing terms will need to be accepted, including [self-archiving policies](https://www.springernature.com/gp/open-research/policies/journal-policies). Those licensing terms will supersede any other terms that the author or any third party may assert apply to any version of the manuscript.

An online order form for reprints of your paper is available at <https://www.nature.com/reprints/author-reprints.html>. All co-authors, authors' institutions and authors' funding

agencies can order reprints using the form appropriate to their geographical region.

With best regards,

[REDACTED]

P.S. Click on the following link if you would like to recommend Nature Human Behaviour to your librarian
<http://www.nature.com/subscriptions/recommend.html#forms>

** Visit the Springer Nature Editorial and Publishing website at http://editorial-jobs.springernature.com?utm_source=ejp_NHumB_email&utm_medium=ejp_NHumB_email&utm_campaign=ejp_NHumB for more information about our career opportunities. If you have any questions please click [here](mailto:editorial.publishing.jobs@springernature.com).

Open Access This Peer Review File is licensed under a Creative Commons Attribution 4.0 International License, which permits use, sharing, adaptation, distribution and reproduction in any medium or format, as long as you give appropriate credit to the original author(s) and the source, provide a link to the Creative Commons license, and indicate if changes were made. In cases where reviewers are anonymous, credit should be given to 'Anonymous Referee' and the source. The images or other third party material in this Peer Review File are included in the article's Creative Commons license, unless

indicated otherwise in a credit line to the material. If material is not included in the article's Creative Commons license and your intended use is not permitted by statutory regulation or exceeds the permitted use, you will need to obtain permission directly from the copyright holder.

Response to the editors and reviewers

We would like to express our gratitude to the Editor and the Reviewers for providing thorough and constructive feedback on our Manuscript. We acknowledge and accept the suggestions made by the Reviewers, which have made our article significantly stronger. We have addressed their concerns and suggestions in detail and used them to further improve the text and argument. Our point-by-point responses can be found below.

Editor's comments

Your manuscript has now been evaluated by 3 reviewers, whose comments are included at the end of this letter. Although the reviewers find your work to be of interest, they also raise some important concerns. We are interested in the possibility of publishing your study in Nature Human Behaviour, but would like to consider your response to these concerns in the form of a revised manuscript before we make a decision on publication.

We thank the Editor and Reviewers for recognizing the significance of the research question and impact of our study and manuscript. We hope to have addressed the constructive criticisms offered by the Reviewers as thoroughly as possible in the following text.

Reviewer #1:

Remarks to the Author:

This is an important paper on agricultural development and animal management in a major region of the Amazon Basin. The maize data are interesting and the muscovy duck data and interpretations are novel and important. I'm not an isotope expert and leave it to those who are to provide an assessment of the isotope data and their interpretations by the authors.

We thank Reviewer #1 for their highly positive comments on the manuscript and its impact.

.....

Reviewer #2:

Remarks to the Author:

This excellent manuscript represents a major and long awaited contribution to Amazonian archaeology and beyond that surely deserves to be published in Nature Human Behaviour.

We thank Reviewer #2 for their praise of our manuscript. We are glad they found it excellent and deserving of publication in Nature Human Behaviour. We thank them for their very constructive suggestions which we have acted upon and believe have helped improve our paper.

Please find below a few minor suggestions for the authors:

l. 21: I would include the number of human skeletal remains analysed.

Added.

l. 39-40: The sentence confuses the spatial extent of crops such as manioc, squash, etc., with the timing of maize. Please clarify the timing of the appearance of these other crops. Iriarte et al. 2021 summarises them.

We have amended this and added a clearer chronology for the different crops.

l. 43: 'Garden cities' is acceptable for the abstract, but in the text, I would define these low-density urban societies in more specific terms and avoid vague terms such as 'Garden cities'.

We have replaced “Garden Cities” for “low-density urban societies”, as requested.

l. 48: The Casarabe culture's architectural elements should be defined more

detailed beyond 'extensive and intricate complexes'. A four-tiered settlement system should be briefly mentioned.

We have added a brief description to lines 52-54.

l. 58: I would include the full list all the plants, including yams, chilli peppers, and palms.

We have now added.

l. 162: Why not compare with some of the most known case studies of the adoption of maize, like the Mississippian ~1000AD, the Belizian Maya (Kenneth et al. 2020), and the other sites in the Amazon? It would be helpful to include a graph displaying all the Amazonian human skeletons' carbon and nitrogen isotope analysis for reference and clear comparisons.

Two new boxplots showing the $\delta^{13}\text{C}$ and $\delta^{15}\text{N}$ values were made (Fig.S3) adding the data from known Amazonian sites (Maracá, Marajó, Ucayali and Xingu) as well as the suggested data from Belize (Kenneth et al. 2020). We thank Reviewer #2 for this suggestion which we agree has provided a nice reference graph for clearer comparison.

l. 176: The reader would like to know if you have capybara in the faunal record and, if so, what the implications are.

Comment added in lines 182-185.

l. 209: Include a citation after previously considered. The role of maize in Amazonian archaeology has a long history of debate, and this paper should summarise at least some of its aspects.

Citation added.

.....

Reviewer #3:

Remarks to the Author:

In "Direct evidence of pre-colonial maize agriculture and animal management in the Bolivian Amazon" the authors report the results of carbon and nitrogen stable

isotopes analysis performed on collagen of human and faunal bone remains from two late Holocene archaeological sites located in the Bolivian Amazon. The study is supported by a remarkable number of samples (70-86 human samples and 60-68 faunal samples) and correct use of established laboratory and statistical techniques. Overall, the results demonstrate that the human diet in these sites heavily relied on C4 plants (likely maize) and tentatively show the potential role that maize played in the diet of Muscovy ducks, the only known domesticated vertebrate in the lowlands of South America. The aim of the study is of high relevance for the interdisciplinary community of scientists studying the intricate human-environment relationships that took place in the Amazon basin during pre-colonial times. However, while the study sheds light on the importance of maize in the diets of pre-colonial people in southwestern Amazonia, the conclusions reached by the authors regarding animal management practices and particularly the feeding of Muscovy ducks seem inadequately supported by the data provided in the article.

We are glad that the Reviewer found our study to be impressive in terms of sample size and methodology. We thank them also for their thorough and constructive comments which we hope to have addressed in the below.

Please, find below major and minor issues that I encountered while reviewing your manuscript.

Major issues

1) The title of the manuscript indicates that it has “direct evidence of pre-colonial maize agriculture” and I think that it is misleading. The results of the study either provide indirect evidence of pre-colonial maize agriculture or direct evidence of pre-colonial maize consumption, given that the stable isotopes analysis was performed on bone collagen and not on agricultural soil samples (see, for example, <https://doi.org/10.1016/j.jas.2010.06.016>). However, it is true that throughout the text the results are interpreted and discussed in terms of dietary patterns and not agricultural practices. For this reason, I would suggest modifying the title to better represent the scope and content of the manuscript.

We appreciate the Reviewer's point here. Based on their affirmation that **“The results of the study either provide indirect evidence of pre-colonial maize agriculture or direct evidence of pre-colonial maize consumption”**, we have changed the title to “Stable isotope evidence for pre-colonial maize agriculture and animal management in the Bolivian Amazon” as it focuses on the method and does not suggest a direct evidence of agriculture.

We also provide the alternative title of “Direct evidence for pre-colonial maize reliance and animal management in the Bolivian Amazon” as our results show **“...direct evidence of pre-colonial maize consumption”**. We leave it to the Editor to decide which is more appropriate.

2) The study discusses potential animal management in the studied sites (see, for example, title and lines 26-28) on the basis of a) the presence of Muscovy duck remains, one of the few animals domesticated in pre-colonial South America; b) the identification of individuals with confinement-related pathologies (lines 178-179); and c) the high values of $\delta^{13}\text{C}$ detected in the bone collagen of Muscovy ducks, which could suggest that they were fed with maize (lines 177-178). However, there are some critical aspects that require further discussion and clarification. First, as shown in Figure 2, rodents, Muscovy ducks and armadillos have a similar $\delta^{13}\text{C}$ range of values when compared to the values in humans, but only the results obtained from capybaras are discussed (lines 173-177) while the ones from armadillos and other rodents are ignored. This is key if the results obtained from the ducks are going to be interpreted as the effect of maize consumption.

Thank you for the important observation. Despite the higher $\delta^{13}\text{C}$ values found in the rodent, riverine and armadillo groups, their representation in the fauna assemblage recovered is considerably small when compared to deer remains (see von den Driesch & Hutterer 2011, tables 1-6). If we take into account the significant size difference between the taxa, where *M. americana* ranges between around 30-40kgs, rodents between around 3 (*D. punctata*) to 10 kg (*A. paca*), armadillos around 4 kgs and eels around 2 kgs, it is evident that the deer is likely to be the main contributor to the animal protein pool.

These points were added to the main text, with a deeper discussion in S1. As for the faunas' dietary sources, the intermediate $\delta^{13}\text{C}$ values found in the riverine taxa and rodents likely reflects the local balance between C_3 and C_4 plants, although maize is not excluded as a potential contributor. As for the armadillos, their stable isotope values vary considerably between a variety of studies. Reasons for this phenomena are still unknown (perhaps due to their generalist diets often including an assortment of invertebrates), but they tend to have more elevated $\delta^{13}\text{C}$ values than other local taxa, even in pre-maize contexts (see Hermenegildo 2009, van der Merwe et al. 2002).

Second, the confinement-related pathologies are only reported in bones from phase 4 in Salvatierra (see pages 361-362 in von den Driesch & Hutterer 2011, which you cite), while the Muscovy duck bones analyzed in this study come from phases 1, 2, and mainly 3.

Due to logistical issues, it was not possible for us to analyse fauna samples from phases 4 and 5, while von den Driesch & Hutterer 2011 could not analyse samples from phases 1 and 2. The confinement related pathologies found play an important role in demonstrating these animal were likely kept alive and tied down. However, this evidence does not imply domestication or large scale management of animals, as raising the younglings from hunted wild animals as pets (sometimes tied up) is a common practice throughout the Amazon (Cormier 2003 <https://doi.org/10.7312/corm12524>; Erickson 2000 <https://doi.org/10.1017/9781108667593.002>; Zuppi 2022 <https://doi.org/10.4000/jsa.21030>). On the other hand, the stable isotope evidence of maize consumption is consistent through all phases analysed, showing that maize consumption in the duck population was common before phase 4 when the pathologies are reported. Therefore, the stable isotope evidence is a more reliable indication of the scope and chronology of muscovy duck management at Casarabe.

Third, file S2 contains the results of 11 Muscovy duck samples, but in the study only the results of 9 samples are included. Interestingly, only the coracoid samples which have lower $\delta^{13}\text{C}$ values were excluded but I could not find the reason neither in the main text nor in S1 and S2.

Thank you for pointing out this inconsistency in the data. These samples had a number of issues indicating they were likely misidentifications. As mentioned before, this

material was not studied by von den Driesch & Hutterer 2011, relying on preliminary identification of limited material. The two coracoid samples analysed were very small (<1g) favouring a potential misidentification, particularly after von den Driesch & Hutterer 2011 identified remains of another duck genus (*Anas* sp.). The samples displayed remarkably similar stable isotope results among themselves, with $\delta^{13}\text{C}$ values diverging by about +6‰ to the rest of the duck population. These values are more in line with the Rodent and Riverine groups, likely reflecting the local balance of C_3 and C_4 sources available in the wild. Considering these were the only coracoid samples analysed, and that these were recovered in different occupation phases and excavation units but show similar isotope values, we concluded these were misidentified remains. For that reason, both were removed from the Bayesian inference analysis in Figure 2. As these could also be wild muscovy duck samples, the values were included in the comparison with archaeological domesticated muscovy duck from Panama (from Sugiyama et al. 2020) presented in Fig. S4 and Tab. S4. This was clarified in S1.

Fourth, the sample size of Muscovy duck is rather low (9-11 bones) compared to the total population of bones recovered in Salvatierra (according to von den Driesch & Hutterer 2011, 89 fragments were recovered in phase 3, 283 in phase 4, and 47 in phase 5) and mostly focused on phase 3, so I wonder if the sampling strategy could have had any effect in the results obtained.

As previously mentioned, there were logistical issues regarding sample availability. We attempted to collect more samples for analysis after we had the data indicative of maize diets but it was impossible at the time. We could not expand the sample size at the time of collection since the most representative phase 4 assemblage was not available so we decided to save the limited duck samples recovered in phases 1, 2 and 3 from a destructive procedure. This small sample size is what leads us to conclude in line 184 that “our data strongly indicate that muscovy ducks were kept and intentionally fed maize” rather than a confirmation they were domesticated, as it was the case in the colonial period (lines 188/9, as mentioned in Denevan 1966).

Minor issues

1) The number of samples does not seem to be consistent throughout the text, something that should be justified. In lines 100-101, 86 human bone remains and 68 faunal remains are reported. In lines 107-112, the total number of faunal remains is 60, while according to file S2 it should sum 68 (or 66, considering that 2 Muscovy duck samples were excluded too). In lines 138-142, the total number of human remains is 70, but why is not 86? Also, in related Table S2, the number of human samples sum 73 instead of 70 or 86.

Thank you for noticing this mistake on our part. The correct values are 86 human remains analysed, 73 of them identified according to ceramic phase. As for the fauna, the total number of results in S2 sums up to 66. The total number in lines 107-112 is indeed 60, considering 2 deer (mentioned in S2), 2 muscovy duck (coracoids) and 2 snake samples were not included in this description. This clarification was added to S2 and we have kept the numbers consistent through the main text.

2) Sugiyama et al. (2020, <https://doi.org/10.1016/j.jaa.2020.101195>) reported similar results and interpretations to yours but using samples from pre-colonial Panamanian sites. I think that the study should be cited and discussed in your manuscript.

Thank you so much for suggesting this publication. The results from this publication were compared with our obtained data and are available in Fig. S4 and Tab. S4. This was added to the main text, lines 193-195.

3) In Figure 2, snakes are excluded in the bottom plot, please explain the reason in the caption and/or somewhere else.

The obtained sample size for snakes ($n=2$) was too small for any relevant interpretations based on Bayesian ellipses, as the analysis progressively loses consistency in sample sizes <10 (see Jackson et al. 2011, <https://doi.org/10.1111/j.1365-2656.2011.01806.x>). Added to caption.

4) Authors should review the manuscript paying careful attention to formatting details. Please, find below some issues that should be solved:

- Line 109. Delete repeated percentage symbols.

Deleted.

- Lines 113-114. To standardize, add comma after the name of the test (Kruskal-Wallis) and a space between the H value and the p value (H(4)=37.13, p<0.001).

Added.

- Lines 120-127. Italicize scientific names in the figure caption.

Corrected.

- Line 143. Replace “:” with “,” after Kruskal-Wallis.

Corrected.

- Line 144. Add name of the test to the parenthesis (ANOVA).

Added.

- Line 233. Replace “Stable” with “stable”.

Corrected.

- Line 461. You might remove “The authors declare no conflicts of interest” from the acknowledgements section, considering that a similar statement is provided in the competing interests section.

Removed.

- Sometimes you refer to supplementary files as, for example, S1 and sometimes as Supplement 1. Please standardize.

Corrected.

- Lines 182-185 and 203-204 in S1. Adjust alignment.

Adjusted.

- Line 183 in S1. Remove the bold font in $\alpha=0.05$.

Removed.

- Line 193 in S1. I recommend describing the acronym SRMs for inexperienced readers (Standard Reference Materials).

Added.

- Line 204 in S1. Please standardize the use of spaces in ME=Mendoza site and SAL=Salvatierra site.

Fixed.